# Massive antibody discovery used to probe structure–function relationships of the essential outer membrane protein LptD

Kelly M Storek[1], Joyce Chan[2], Rajesh Vij[3], Nancy Chiang[3], Zhonghua Lin[3], Jack Bevers III[3], Christopher M Koth[4], Jean-Michel Vernes[2], Y Gloria Meng[2], JianPing Yin[4], Heidi Wallweber[4], Olivier Dalmas[4], Stephanie Shriver[5], Christine Tam[5], Kellen Schneider[3], Dhaya Seshasayee[3], Gerald Nakamura[3], Peter A Smith[1], Jian Payandeh[4]*, James T Koerber[3]*, Laetitia Comps-Agrar[2]*, Steven T Rutherford[1]*

[1]Department of Infectious Diseases, Genentech, Inc, South San Francisco, United States; [2]Department of Biochemical and Cellular Pharmacology, Genentech, Inc, South San Francisco, United States; [3]Department of Antibody Engineering, Genentech, Inc, South San Francisco, United States; [4]Department of Structural Biology, Genentech, Inc, South San Francisco, United States; [5]Department of Biomolecular Resources, Genentech, Inc, South San Francisco, United States

*For correspondence:
payandeh.jian@gene.com (JP);
koerber.james@gene.com (JTK);
compsagrar.laetitia@gene.com
(LC-A);
rutherford.steven@gene.com
(STR)

**Abstract** Outer membrane proteins (OMPs) in Gram-negative bacteria dictate permeability of metabolites, antibiotics, and toxins. Elucidating the structure-function relationships governing OMPs within native membrane environments remains challenging. We constructed a diverse library of >3000 monoclonal antibodies to assess the roles of extracellular loops (ECLs) in LptD, an essential OMP that inserts lipopolysaccharide into the outer membrane of *Escherichia coli*. Epitope binning and mapping experiments with LptD-loop-deletion mutants demonstrated that 7 of the 13 ECLs are targeted by antibodies. Only ECLs inaccessible to antibodies were required for the structure or function of LptD. Our results suggest that antibody-accessible loops evolved to protect key extracellular regions of LptD, but are themselves dispensable. Supporting this hypothesis, no α-LptD antibody interfered with essential functions of LptD. Our experimental workflow enables structure-function studies of OMPs in native cellular environments, provides unexpected insight into LptD, and presents a method to assess the therapeutic potential of antibody targeting.
DOI: https://doi.org/10.7554/eLife.46258.001

## Introduction

The outer membrane (OM) of Gram-negative bacteria is a permeability barrier to antibiotics and other cytotoxic agents, such as detergents (*Nikaido, 2003*). A key feature of the OM is its distinctive asymmetrical bilayer populated with lipopolysaccharide (LPS) in the outer leaflet and phospholipids in the inner leaflet (*Funahara and Nikaido, 1980*; *Kamio and Nikaido, 1976*). Lateral interactions between LPS molecules mediated by divalent cations on the surface and packing of hydrocarbon chains in the membrane impede the passage of both hydrophilic and large hydrophobic molecules (*Nikaido, 2003*). LPS is composed of a conserved lipid A molecule, a core oligosaccharide, and a variable O-antigen glycan. Lipid A is synthesized in the cytoplasm where the core oligosaccharide is attached to form core-LPS (*Whitfield and Trent, 2014*). This molecule is flipped into the periplasmic

**eLife digest** The overuse and misuse of antibiotics has led to the rise of multi-drug resistant bacteria which threaten global public health. Antibiotics interfere with essential processes in bacteria so they are unable to divide or survive, but over time, the microbes have found ways to become immune to the drugs. New antibiotics are now desperately needed.

Gram-negative bacteria are wrapped in an outer membrane made of large molecules called lipopolysaccharides. This structure is an extra barrier to molecules (such as drugs) that try to enter the cell, but it could also hold new targets for antibiotics to exploit.

A protein called LptD is embedded in the outer membrane, where it inserts new lipopolysaccharides. It is critical for bacteria to grow and survive, and is a relatively new potential target for antibiotic development. The protein has a number of 'extracellular loops' that extend into the environment, but their roles in the structure and the activity of LptD are still largely unknown. This is partly due to a lack of tools to investigate these elements.

In response, Storek et al. built a library of over 3,000 custom antibodies, which are small Y-shaped proteins that can each recognise a specific portion in one of the extracellular loops and potentially incapacitate LptD. The antibodies were used to target LptD in its native environment, when it is embedded in the bacteria. In parallel, mutant bacteria were created in which the loops were genetically removed one by one to assess their importance for LptD activity.

The experiments revealed that although the antibodies could target most extracellular loops, they could not target the few loops that were essential for LptD to work properly. This suggests that antibody-accessible loops are expendable and that these structures could serve to shield other regions of LptD which are critical for survival.

The findings will help to prioritise research that develops other approaches to inhibit LptD. Finally, the antibody workflow designed by Storek et al. can serve as a road map to study other membrane proteins in their native cellular environment.

DOI: https://doi.org/10.7554/eLife.46258.002

space by the inner membrane ATPase MsbA (*Ho et al., 2018*; *Mi et al., 2017*; *Zhou et al., 1998*). The O-antigen is appended onto core-LPS in the periplasm and LPS is transported to the outer leaflet of the OM by the LPS transport (Lpt) system (*Abeyrathne et al., 2005*; *Han et al., 2012*; *Okuda et al., 2016*).

In *E. coli*, all 7 components of the Lpt pathway, LptABCDEFG, are essential and shuttle LPS molecules from the IM across the periplasm to the OM (*Okuda et al., 2016*; *Okuda et al., 2012*; *Ruiz et al., 2008*; *Wu et al., 2006*). LptD is a β-barrel outer membrane protein (OMP) responsible for the final unidirectional insertion of LPS into the OM outer leaflet (*Botos et al., 2016*; *Gu et al., 2015*; *Li et al., 2015*). The LptD β-barrel is comprised of 26 anti-parallel β-strands with a proposed lateral gate formed where the first and last β-strands interact (*Dong et al., 2014*; *Qiao et al., 2014*). Like other Gram-negative β-barrel OMPs, LptD possess long extracellular, environmentally-exposed loops and short periplasmic loops connecting adjacent strands (*Franklin and Slusky, 2018*; *Rollauer et al., 2015*; *Schulz, 2002*). The roles of the extracellular loops (ECLs) in folding or activity of LptD have not been determined. LptD forms an obligate interaction with the OM lipoprotein LptE (*Chimalakonda et al., 2011*; *Freinkman et al., 2011*). An N-terminal jelly roll domain is located on the periplasmic side of the β-barrel below the lateral gate. Structural, biochemical, and genetic analyses suggest that the hydrophobic lipid A portion of LPS is embedded in the hydrophobic jelly roll domain. The presumed model for LPS transport across the OM posits that lipid A is inserted into the hydrophobic OM through a lateral gate of LptD (*Botos et al., 2016*; *Gu et al., 2015*; *Li et al., 2015*). By an unknown mechanism, the polar sugars of the LPS core oligosaccharide and O-antigen are presumably moved through the LptD β-barrel lumen and flipped to the exterior of the cell.

Due to its essential role in OM biogenesis, high conservation among Gram-negative bacteria, and exposure to the extracellular environment, LptD represents a potential new antibacterial target. Genetic experiments show that crosslinking LPS to LptD in the jelly roll domain or the lateral gate, or introduction of disulfide bonds that disrupt the proposed LPS path through LptD, inhibit growth (*Gu et al., 2015*; *Li et al., 2015*). Indeed, two antibacterial peptides targeting LptD have been

described, both presumably targeting the periplasmic jelly roll domain (*Srinivas et al., 2010*; *Vetterli et al., 2018*). Critical roles for ECLs in OMP function have been described (*Browning et al., 2013*; *Rigel et al., 2013*). For example, removal of the LptD extracellular loop 4 (L4), *imp*4213, is known to disrupt the OM barrier leading to sensitization to antibiotics (*Sampson et al., 1989*). However, the roles of the extracellular LptD loops remain largely underexplored due to a lack of robust tools to systematically assess structure-function relationships in the context of a native OM.

Monoclonal antibodies (mAbs) represent useful tools to probe protein structure and function because they typically have high target specificity and affinity. Accordingly, here we pursued four diverse strategies to produce thousands of mAbs that bound to LptD. The breadth and depth of mAb discovery that we report here is unusual, especially for an integral membrane protein, and also takes advantage of an efficient and novel targeted boost-and-sort α-OMP discovery approach recently reported for another OMP, BamA (*Vij et al., 2018*). Characterization of the α-LptD mAbs discovered herein allowed us to map the environmentally-exposed surfaces of LptD and to systematically explore the structure-function relationship of the large ECLs of this essential OMP within its native OM environment. Through the construction and characterization of LptD loop deletion strains, we identified non-exposed loops that are critical for LptD function under defined conditions, which are all inaccessible to antibody interference. Our data provide unique insights into the structure and function of LptD and suggest that non-antibody approaches for interfering with the function of this essential OMP in *E. coli* should be prioritized to develop novel antibiotics.

## Results

### Identifying α-LptD mAbs that bind to extracellular LptD loops

To probe the functional relevance of the ECLs of the essential *E. coli* OMP LptD, we set out to discover α-LptD antibodies to these surface-exposed structures. We generated a diverse mAb library against *E. coli* LptD by running four independent discovery campaigns. Specifically, we immunized (1) rats with cyclic and linear peptides derived from extracellular LptD loop sequences, (2) rats and (3) mice with purified LptDE protein in detergent (n-octyl-β-D-glucopyranoside) and non-detergent polymer (amphipol A8-35), and (4) rats with a targeted boost-and-sort strategy that uses whole bacterial cell immunizations followed by immune boosts with purified LptDE protein (*Figure 1A* and *Figure 1B*). Within each campaign, the adjuvants and boosting strategies were varied to increase potential antibody diversity (see Materials and methods for details). Cumulatively, the discovery campaigns generated more than 7000 hybridomas and more than 3,000 ELISA+ LptD-specific mAbs (*Figure 1C*). In total, 774 ELISA+ α-LptD mAbs from the peptide and protein rat immunizations, 952 ELISA+ α-LptD mAbs from the mouse immunizations, and 1494 α-LptD mAbs from the targeted boost-and-sort immunizations were purified (*Figure 1C*).

To explore epitope coverage and accessibility of the α-LptD ELISA+ mAbs from three of the antibody discovery campaigns, we first measured their extracellular binding by a fluorescence-assisted cell sorting (FACS)-based assay to two strains of *E. coli* (*Figure 2A*). The first strain was a laboratory *E. coli* strain, K-12, that produces an intact core-LPS but no O-antigen. The second strain was an *E. coli ΔwaaD* mutant, which produces a truncated form of LPS on the cell surface (*Kneidinger et al., 2002*). This truncation of LPS increases the exposed epitopes accessible to mAbs (*Bentley and Klebba, 1988*; *Storek et al., 2018*). Depending on the immunization campaign, 9–30% of the ELISA+ α-LptD mAbs were FACS+ on *E. coli ΔwaaD*, while <1% of the same antibodies were FACS+ on *E. coli* K-12 (*Figure 2A*). The *E. coli* K-12 FACS+ α-LptD mAbs were a subset of all the FACS+ mAbs, binding equally well to *E. coli* K-12 and *E. coli ΔwaaD* cells (*Figure 2—figure supplement 1*), and were discovered in both mouse and rat immunization campaigns. These results reinforce the finding that LPS presents a significant barrier to accessing surface epitopes on *E. coli* (*Bentley and Klebba, 1988*; *Storek et al., 2018*) and suggest that the ECLs of OMPs may be masked or intimately associated with the core sugars of the LPS.

The diversity of epitopes bound by α-LptD mAbs in our library was next assessed by monitoring binding to LptD from different species. ELISAs were performed with 134 FACS+ and 233 FACS- α-LptD mAbs using purified LptDE protein from two closely related Enterobacteriaceae species, *Klebsiella pneumoniae* and *Enterobacter cloacae*, which share 82% and 84% overall protein identity and 71% and 76% identify between the ECLs only when compared with *E. coli* LptD, respectively

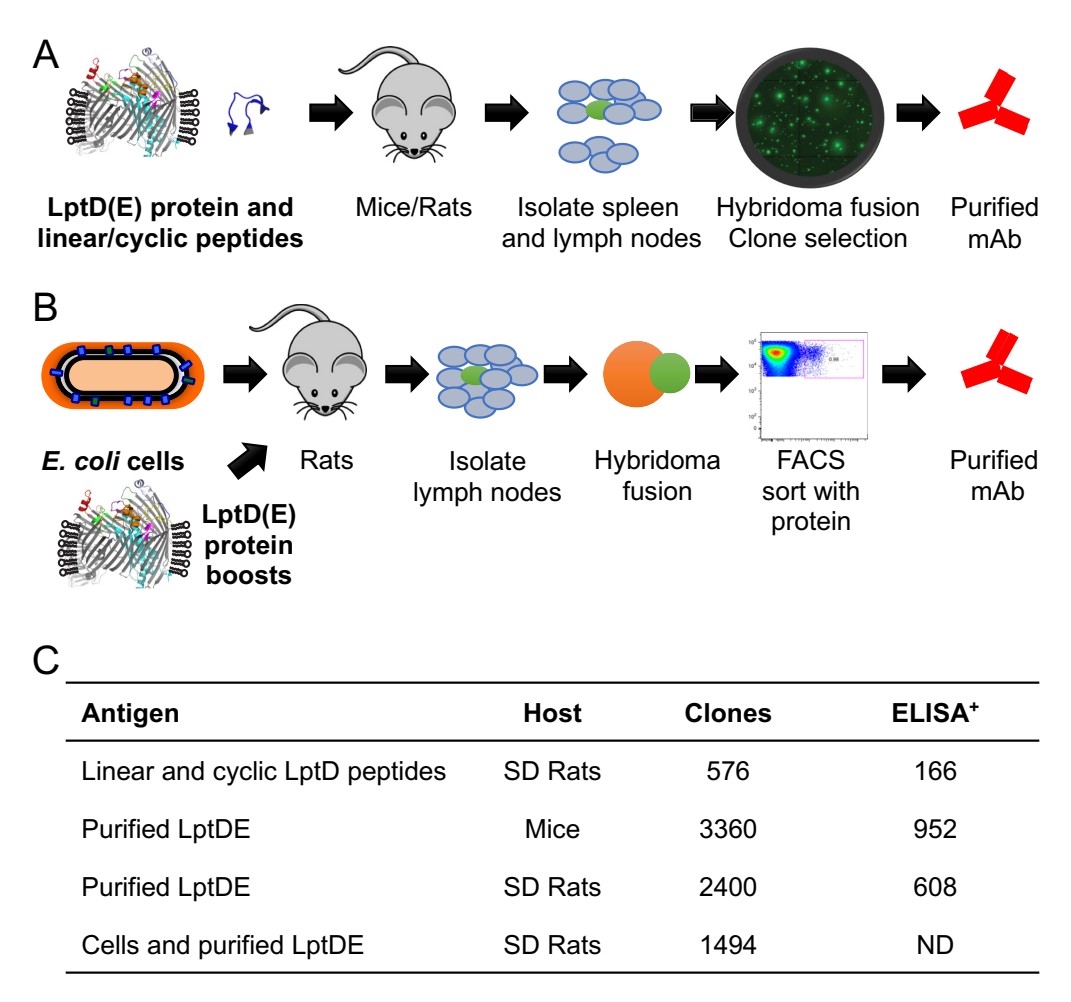

**Figure 1.** Immunization schematics and α-LptD mAb library characteristics. (**A**) Cartoon of LptD immunization campaigns with purified *E. coli* LptDE protein, LptD cyclic peptides, and linear peptides. 10 rounds of protein or peptide injections were performed. Clones were selected from hybridoma fusions. (**B**) Cartoon of targeted boost-and-sort strategy in which SD rats were initially primed with *E. coli* K-12 bacteria, followed by two boosts with the recombinant LptDE protein reconstituted in amphipol matrix. Cell hybridoma fusions were sorted with fluorescently-labeled LptDE to enrich LptDE[+] hybridomas. (**C**) Overview of the individual immunization campaigns. The total number of hybridoma clones isolated and binding to LptDE by ELISA were determined for each campaign. ELISA positive antibodies had a signal 3x above background. ELISA measurements were not determined (ND) for the hybridoma clones from the boost-and-sort approach.

DOI: https://doi.org/10.7554/eLife.46258.003

(*Figure 2B* and *Figure 2—figure supplement 2A*). The majority of FACS[+] (61%) and FACS[-] (69%) α-LptD mAbs bound only *E. coli* LptDE by ELISA (*Figure 2C*). This result was not surprising considering that all protein immunization campaigns utilized only purified *E. coli* LptDE protein. Analysis of the surface-exposed portion of LptD predicted from experimental x-ray crystallographic structural models identified regions of >5 amino acids that were conserved among all three Enterobacteriaceae species, the size of a typical hot spot of residues that contribute to antibody binding (*Stave and Lindpaintner, 2013*) (*Figure 2B* and *Figure 2—figure supplement 2B*). Indeed, we identified 10 (7%) α-LptD mAbs that bound to LptDE from all three species and 42 that bound either *E. coli* plus *K. pneumoniae* (22%) or *E. coli* plus *E. cloacae* (9%) LptDE (*Figure 2C*). Thus, our immunization campaigns using *E. coli* LptD-derived peptides and *E. coli* LptDE protein yielded diverse antibodies that bound to both unique and conserved extracellular accessible (i.e., FACS[+]) and extracellular-inaccessible or periplasmic (i.e., FACS[-]) epitopes.

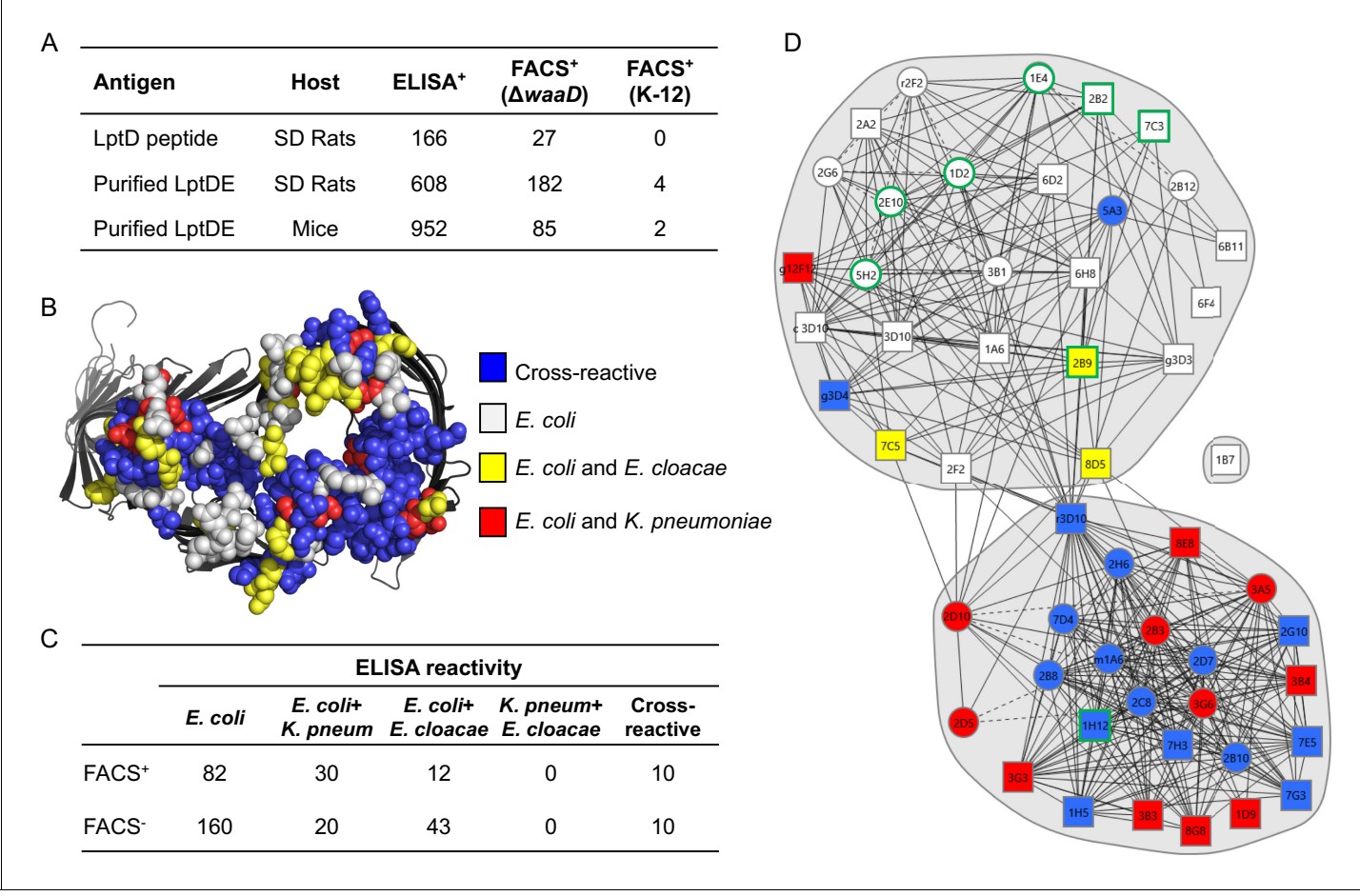

**Figure 2.** Whole cell FACS binding and epitope binning of α-LptD mAbs. (A) LptD mAb campaign summary of ELISA⁺ LptDE mAbs tested for surface binding on *E. coli* Δ*waaD* and *E. coli* K-12 by FACS assay. Antibodies were scored FACS⁺ if the MFI was 2x above an isotype control. (B) Amino acids comprising the 13 ECLs of LptD are highlighted on the structure as spheres. The spheres are color-coded based on amino acid conservation between different sequence comparisons as indicated in the key. Structure is of *Shigella flexneri* LptDE with LptE removed rendered in PyMol (PDB 4Q35 [*Qiao et al., 2014*]). (C) 134 FACS⁺ and 233 FACS⁻ antibodies were characterized for cross-reactive binding to purified LptDE from two closely-related Enterobacteriaceae species: *Klebsiella pneumoniae* (*K. pneum*) and *Enterobacter cloacae* (*E. cloacae*) by ELISA. ELISA positive antibodies had a signal 3x above background. (D) 52 FACS⁺ α-LptD mAbs were characterized for epitope binning patterns using a high-throughput SPR-based method to determine pairwise mAb binding competition. Antibodies are color-coded based on ELISA cross-reactivity profiles as indicated in the key. Circle designates data obtained from both capture and detection antibodies. Square designates a single capture or detection data point. Characterization of all antibodies described in *Figure 4—figure supplement 1* and those with green-outlined squares or circles are shown in *Figure 4B*.
DOI: https://doi.org/10.7554/eLife.46258.004

The following figure supplements are available for figure 2:

**Figure supplement 1.** α-LptD mAb binding to *E. coli* K-12 and *E. coli* Δ*waaD* strains by FACS analysis.
DOI: https://doi.org/10.7554/eLife.46258.005
**Figure supplement 2.** LptD ECL conservation among different Enterobacteriaceae species.
DOI: https://doi.org/10.7554/eLife.46258.006

The differences in cross-species reactivity of the α-LptD mAbs suggested that the FACS⁺ antibodies bound multiple surface-exposed LptD epitopes. To determine if these antibodies bound unique extracellular epitopes, we tested α-LptD mAbs with the highest FACS⁺ signal for their abilities to compete with each other for binding to purified *E. coli* LptDE. Every possible FACS⁺ pair of α-LptD mAbs was screened for the ability to either bind LptDE protein simultaneously (i.e., different epitopes) or to compete for binding (i.e., same or nearby epitopes) using a SPR-based binding assay (*Abdiche et al., 2017*). From these results, we constructed a surface epitope map for the FACS⁺ α-LptD mAbs (*Figure 2D*). We identified two major epitope bins and a possible third bin with limited

overlap (*Figure 2D*). When the species cross-reactivity data were considered, most three-species cross-reactive α-LptD mAbs were tightly clustered, while those that bound only *E. coli* LptDE formed a distinct cluster (*Figure 2D*). There were some outliers to these trends with the dual-species binders and *E. coli*-only binders binning with the cross-reactive mAbs suggesting distinct, but overlapping epitopes. Thus, the α-LptD mAbs we discovered covered multiple distinct and overlapping extracellular LptD epitopes.

## Effects of removing extracellular LptD loops on *E. coli* growth

In order to map the extracellular epitopes of the α-LptD mAbs in a native asymmetrical OM environment, we first generated and characterized a panel of *lptD* mutants each lacking one ECL. This strain panel also allowed us to assess the essentiality of each ECL in strains with different OM compositions. We constructed 13 *lptD* mutants, each lacking the region encoding one of the 13 ECLs

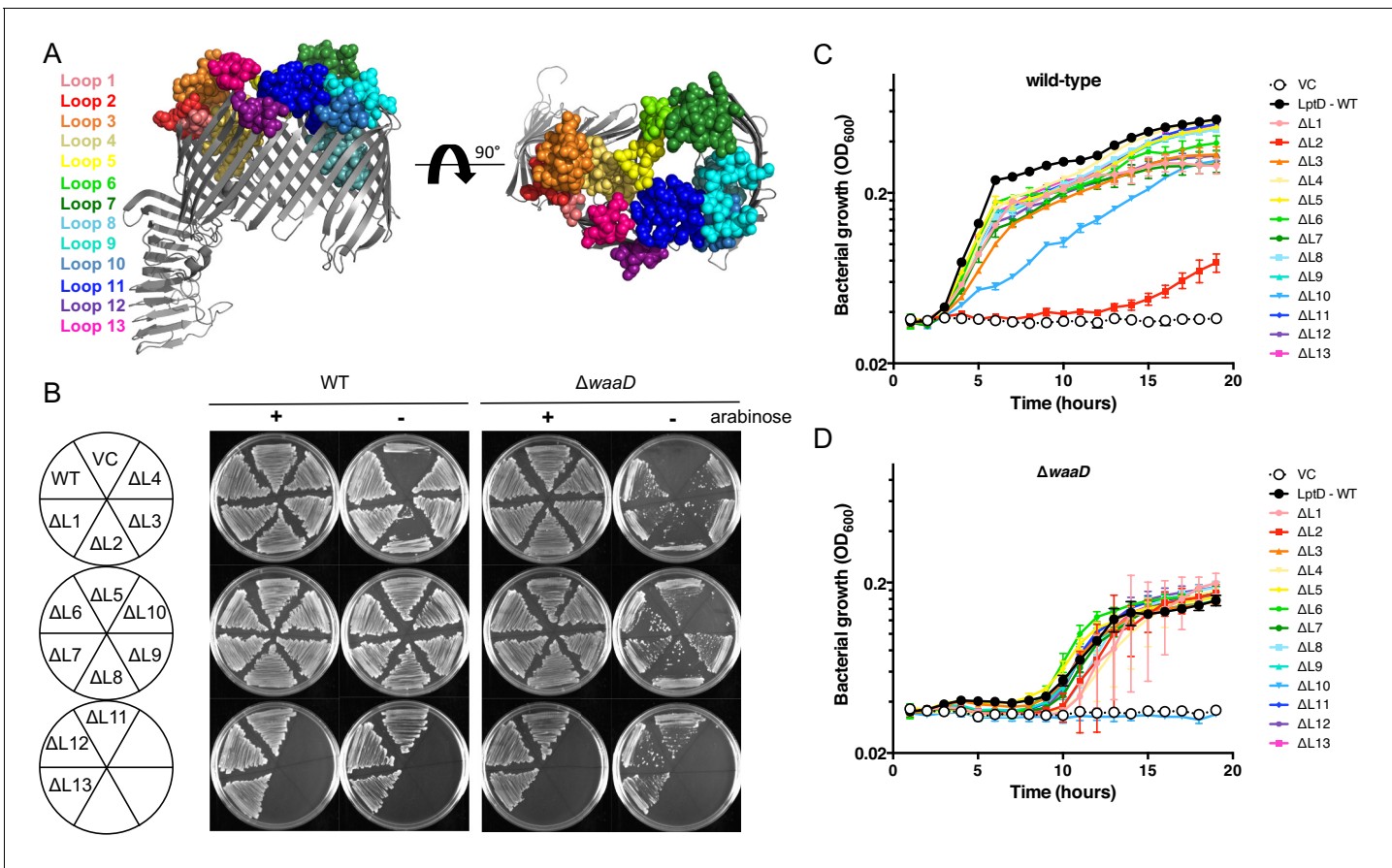

**Figure 3.** LptD ECLs are dispensable for *E. coli* growth. (A) The ECL boundaries of LptD are highlighted as spheres and color-coded as indicated. Sequence boundaries are indicated *Figure 2—figure supplement 2A*. Structure based on *S. flexneri* LptDE with LptE removed rendered in PyMol (PDB 4Q35 [*Qiao et al., 2014*]). (B) LptD loop mutants were expressed on a plasmid, pLMG18gm, in a conditional *lptD* mutant strain in which the chromosomal copy of wild-type *lptD* is under arabinose control. Mutants were streaked onto LB gentamicin agar supplemented with or without 0.2% arabinose and imaged after overnight growth. Representative image of three biological triplicates is shown. Growth was determined in both the *E. coli* wild-type and *E. coli* Δ*waaD* strain backgrounds. Vector control (VC) does not encode a copy of *lptD* and requires arabinose induction of the chromosomal *lptD* copy to grow. Growth curves of *lptD* loop mutants expressed in *E. coli* (C) K-12 and (D) Δ*waaD* strain backgrounds are shown. Strains were grown from a starting inoculum of $OD_{600}$ 0.001 in LB gentamicin without arabinose and monitored for bacterial growth by measuring $OD_{600}$ over 15 hr. Means and standard deviations from biological triplicates are plotted. Statistical analysis of growth rates for each curve are shown in *Supplementary file 1*.

DOI: https://doi.org/10.7554/eLife.46258.007

The following figure supplement is available for figure 3:

**Figure supplement 1.** Expression of *lptD* loop mutants does not affect growth of *E. coli* expressing a wild-type copy of *lptD*.

DOI: https://doi.org/10.7554/eLife.46258.008

(*Figure 3A*). These loop deletions were based on available x-ray crystal structures of LptD and were constructed as complete loop deletions without replacing the removed sequence. In the case of loop 4 (L4), which encompassed the classical *imp4213* allele (*Sampson et al., 1989*) and in the case of loop 8, which is an extensive loop that interacts with LptE in the LptD lumen (*Freinkman et al., 2011*), we only removed a portion of the loop. Each *lptD* loop mutant was expressed in a *lptD*-conditional *E. coli* strain. In the presence of arabinose, which induces a chromosomal wild-type *lptD*, all strains grew (*Figure 3B*). In the absence of arabinose, all *lptD* loop mutants supported growth in the K-12 strain with the exception of loop 2 (L2) removal, which showed reduced growth (*Figure 3B*). In all cases when growth was observed, the levels of the LptD loop deletion protein produced were similar to the protein level of wild-type LptD (*Figure 3—figure supplement 1A*). Thus, LptD is highly tolerable to genetic manipulation of its ECLs.

When the *lptD* loop mutants were expressed in an *lptD*-conditional *E. coli* ΔwaaD strain, which produces LPS with a truncated core oligosaccharide, the observed loop requirements were different. In the *E. coli* ΔwaaD background, LptD lacking either loop 2 or loop 10 were unable to support bacterial growth on solid growth agar (*Figure 3B*). The inability of LptD lacking L10 (LptDΔL10) to support growth of the *E. coli* ΔwaaD strain suggested that this ECL is important for LptD folding or activity in this strain. In contrast, the LptDΔL10 variant was produced and did support growth of *E. coli* K-12 (*Figure 3B* and *Figure 3—figure supplement 1A*), indicating that LptDΔL10 was folded and sufficiently functional in the strain producing core-LPS but not the minimal LPS of *E. coli* ΔwaaD strain.

To determine if the other LptD loops had differential activities in these two strains, we monitored the growth of all of the LptD loop variants in both the *E. coli* K-12 and *E. coli* ΔwaaD strain backgrounds over time in liquid growth media. In the presence of arabinose, which induces a chromosomal wild-type *lptD*, all strains had similar growth patterns (*Figure 3—figure supplement 1C*). In the absence of arabinose, the *E. coli* K-12 strain producing LptDΔL10 grew, but at a decreased rate, and the strain with LptDΔL2 exhibited a dramatically prolonged lag phase and severe growth phenotype (*Figure 3C* and *Supplementary file 1*). In contrast, *E. coli* ΔwaaD expressing LptDΔL10 did not grow at all, but expression of LptDΔL2 supported growth in liquid media (*Figure 3D* and *Supplementary file 1*). Thus, L2 and L10 are important for LptD folding, activity, interaction with the β-barrel assembly machinery component BamA, or a combination of these. Also, the LptDΔL10 loop deletion mutant was better tolerated in *E. coli* K-12 compared to the LPS-truncated ΔwaaD strain while in liquid growth media the LptDΔL2 mutant is better tolerated in the *E. coli* ΔwaaD strain. This result is consistent with the synthetic lethality of other mutations that affect OM biogenesis, specifically *bam101*, ΔbamB, ΔsurA, and ΔlpxL, in LPS-truncated *E. coli* strain backgrounds (*Klein et al., 2009*; *Storek et al., 2019*) highlighting the redundancy that exists to maintain this important barrier structure.

The *lptD*ΔL4 mutant used in this study encompassed the *imp4213* allele. This particular loop mutant is viable but increases membrane permeability and sensitivity to OM-excluded antibiotics (*Sampson et al., 1989*). We tested whether the strains producing other LptD loop mutants had increased membrane permeability by measuring sensitivities to rifampicin and vancomycin, two antibiotics excluded by the Gram-negative bacteria OM. The minimal inhibitory concentrations (MICs) of both rifampicin and vancomycin were lower for strains producing LptDΔL4, as expected, and also for strains producing LptDΔL2, LptDΔL8, and LptDΔL10 indicating an OM defect (*Table 1*). Increases in OM permeability were observed in both *E. coli* K-12 and *E. coli* ΔwaaD strain backgrounds. These results suggest that extracellular L2, L4, L8 and L10 are important for LptD function and potentially influence LptD folding, interaction with BamA, or activity.

## Mapping surface loop binding sites of α-LptD antibodies

We utilized our *lptD* loop deletion panel to map the α-LptD mAb surface coverage in a native *E. coli* membrane environment. We characterized binding of 52 FACS⁺ α-LptD mAbs to the *E. coli* ΔwaaD loop deletion panel. In the absence of arabinose driving expression of the chromosomal wild-type *lptD*, production of each mutant LptD was detected and growth of the *E. coli* ΔwaaD conditional *lptD* strain was supported by all LptD loop deletion mutants with the exception of L10, so this construct was not tested in this assay (*Figure 3B* and *Figure 3—figure supplement 1B*). The mAbs in the panel were confirmed to bind to an *E. coli* ΔwaaD strain expressing wild-type *lptD* (*Figure 4A*). We expected that if an extracellular LptD loop was critical for binding, removing the loop would

**Table 1.** Removal of extracellular LptD loops sensitizes *E. coli* to OM-excluded antibiotics.

| | MIC (μg/ml)* | | | |
| | Vancomycin | | Rifampicin | |
| LptD† | WT‡ | ΔwaaD§ | WT | ΔwaaD |
|---|---|---|---|---|
| WT | 128 | 64 | 8 | 0.125 |
| ΔL1 | 128 | 64 | 4 | 0.125 |
| ΔL2 | 8 | 4 | 0.25 | 0.0625 |
| ΔL3 | 128 | 16 | 4 | 0.125 |
| ΔL4 | 16 | 1 | 0.25 | 0.0156 |
| ΔL5 | 128 | 64 | 8 | 0.125 |
| ΔL6 | 128 | 64 | 2 | 0.0625 |
| ΔL7 | 128 | 64 | 4 | 0.125 |
| ΔL8 | 16 | 4 | 0.125 | 0.0625 |
| ΔL9 | 128 | 128 | 4 | 0.125 |
| ΔL10 | 64 | NG | 2 | NG |
| ΔL11 | 128 | 64 | 8 | 0.125 |
| ΔL12 | 128 | 64 | 8 | 0.125 |
| ΔL13 | 128 | 64 | 8 | 0.125 |

*Minimum inhibitory concentration (MIC) is the lowest concentration of antibiotic that completely inhibits bacterial growth.

†Conditional *lptD E. coli* strains carry an arabinose-inducible wild-type *lptD* and a plasmid-encoded copy of *lptD* with the indicated loop deletions (as indicated in **Figure 3A**). Only the plasmid copy of *lptD* is expressed in the absence of arabinose.

‡The wild-type (WT) strain is a conditional *E. coli* K-12 with a chromosomal arabinose-inducible *lptD*.

§The *ΔwaaD* strain is a conditional *E. coli ΔwaaD* mutant with a chromosomal arabinose-inducible *lptD*.

DOI: https://doi.org/10.7554/eLife.46258.009

reduce α-LptD mAb binding. The lack of binding by any particular mAb to a LptD loop mutant could be due to removal of the mAb binding site, in part or in full, or a structural change that alters the mAb binding site when a loop is removed. The majority of the α-LptD mAbs could be categorized into one of 7 general binding patterns based on the mutants they were unable to bind with multiple representatives in each grouping: (1) Loop 4 (L4), (2) Loop 6/7 (L6/L7), (3) Loop 6/7/8 (L6/L7/L8), (4) Loop 8/9 (L8/L9), (5) Loop 9 (L9), (6) Loop 11 (L11), and (7) Loop 13 (L13) (**Figure 4B** and **Figure 4—figure supplement 1**). Thus, our immunization campaigns produced antibodies that allow systematic probing of accessible ELCs around the LptD structure.

In some cases, for example, a representative *E. coli* K-12 FACS⁺ α-LptD mAb (**Figure 2—figure supplement 1B**), no loop mutant dramatically altered antibody binding. This pattern suggests a surface-exposed epitope was present that was not altered in these particular loop mutants or the mAb bound a composite epitope that was not sufficiently disrupted by loss of a portion of the epitope encompassed by one individual loop deletion (**Figure 4B**). This could be especially true for L8 as the deletion encompassed only part of the loop that, based on structural models, is likely to occupy the LptD lumen while additional exposed portions were left unchanged (**Figure 4C**).

For every loop deletion, at least one FACS⁺ α-LptD mAb had reduced cell binding by >90%. The notable exceptions were L1, L2, L5 and L12, but mAbs could be found that exhibited >80% decreased binding for L1, L5 and L12. Comparing these data with the species cross-reactivity and in vitro binding competition epitope map provided a complete picture of the mAb accessible surface of LptD. Specifically, a majority of the two- and three-species cross-reactive α-LptD mAbs were dependent upon L8, L9, or both for binding (**Figure 2B**, **Figure 2D**, **Figure 2—figure supplement 2**, **Figure 4B**, and **Figure 4—figure supplement 1**). The cross-reactive mAbs 5A3 and 3D4 exhibited slightly different binding requirements, L6/L8/L9/L11 and L7/L8, respectively, consistent with these mAbs recognizing different epitopes in our in vitro binning experiment (**Figure 2D**). Additionally, *E.*

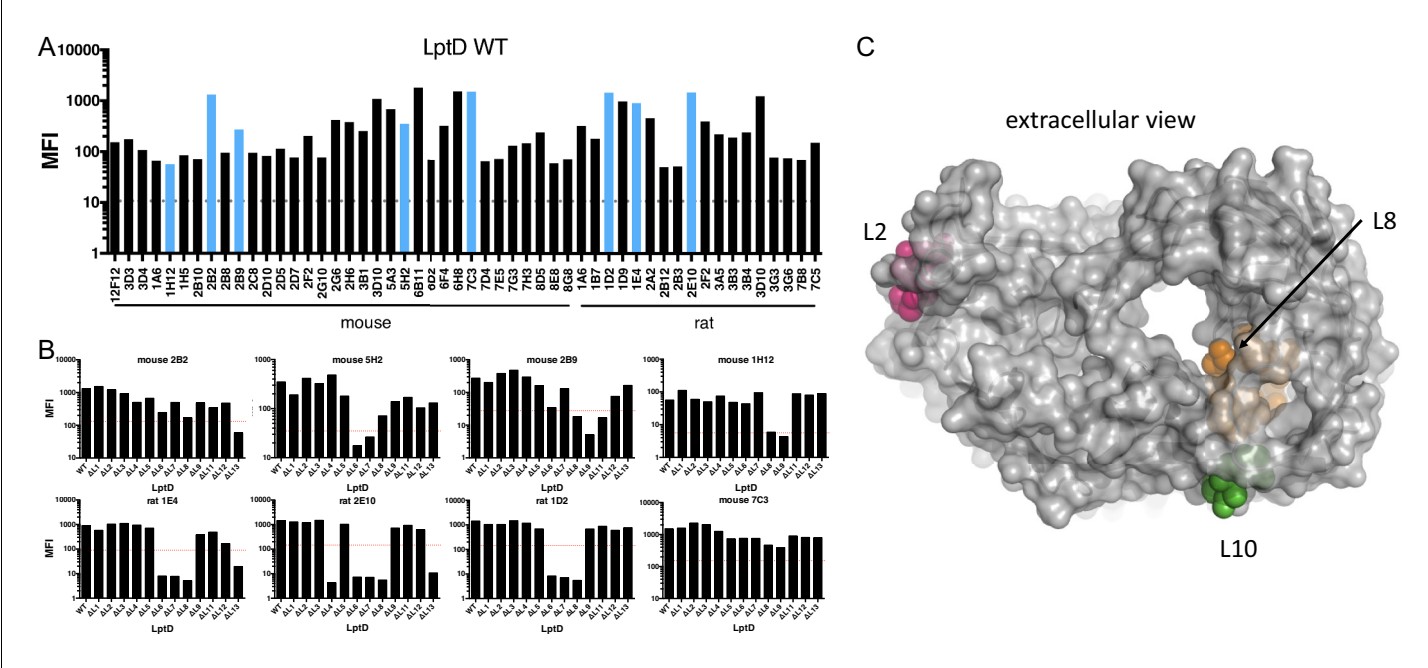

**Figure 4.** Mapping α-LptD mAb binding using LptD loop variants. (**A**) 52 FACS⁺ LptD mAbs were screened for FACS binding to *E. coli* ΔwaaD expressing wild-type *lptD*. The mean fluorescent intensities (MFIs) are plotted. The blue bars highlight the eight representative mAbs shown below in 4B. Antibodies originating from different hosts are labeled. The dotted line is the MFI of an LptD FACS⁻ antibody. (**B**) FACS analysis of 8 representative α-LptD mAbs for binding to *E. coli* ΔwaaD expressing wild-type LptD and the 12 viable loop deletions. Loop 10 could not be assessed (see text). The MFIs above background are plotted. The red dashed lines represent a descriptive MFI level that is 10% that of the WT control. The remaining antibodies from (**A**) are shown in *Figure 4—figure supplement 1*. (**C**) Top view of LptD structure from *E. coli* LptDE with LptE removed highlighting extracellular L2 (pink), L8 (orange), and L10 (green) rendered in PyMol (PDB 4RHB [*Malojcic et al., 2015*]).

DOI: https://doi.org/10.7554/eLife.46258.010

The following figure supplements are available for figure 4:

**Figure supplement 1.** Comprehensive FACS analysis on all *lptD* loop mutants.
DOI: https://doi.org/10.7554/eLife.46258.011

**Figure supplement 2.** Functional analysis of α-LptD antibodies.
DOI: https://doi.org/10.7554/eLife.46258.012

*coli*-specific α-LptD mAbs, but not the three-species cross-reactive mAbs, were defective for binding upon removal of L3, L7, or L13 (*Figure 2B*, *Figure 2D*, *Figure 2—figure supplement 2*, *Figure 4B*, and *Figure 4—figure supplement 1*). These α-LptD mAbs and knowledge of their binding sites are valuable tools for probing the accessible LptD surface and suggest that LptD can tolerate extensive alteration to its extracellular surface with little or no effect on folding or function.

Because distinct antibody discovery approaches were utilized, we were also able to evaluate the potential correlation of LptD targeting with regard to the immunization campaign. Partitioning the mAbs with respect to the host animal used for immunization highlighted a trend for L6, L7, and L8 to be critical from the rat immunization campaign, while mAbs from the mouse immunization campaign tended to yield FACS⁺ antibodies that required L8 and L9. Because these animals are likely not tolerized to the foreign bacterial LptD protein, this difference could reflect differences in the types of binding sites presented for rat versus mouse antibodies. Thus, even within these immunization campaigns, diverse binding profiles to the loop mutants were seen (*Figure 4B* and *Figure 4—figure supplement 1*).

## LptD ECLs critical for function are not accessible

A potential application of antibodies raised against extracellularly-exposed bacterial proteins is as direct-acting antibacterial molecules (*LaRocca et al., 2009*; *Storek et al., 2018*). Disruption of L2 or L10 resulted in growth defects for *E. coli* ΔwaaD (*Figure 3B*), however, we did not identify any α-

LptD mAbs that bound L2, and both L2 and L10 appear to be inaccessible in available structural models of LptD (*Figure 4C*) (*Botos et al., 2016*; *Dong et al., 2014*; *Qiao et al., 2014*). Indeed, when we screened our entire antibody catalog using growth inhibition as a readout for mAb activity, we did not identify any α-LptD mAbs able to inhibit growth of *E. coli* K-12 or *E. coli* ΔwaaD by >50% (*Table 2* and *Figure 4—figure supplement 2*). Combined with our LptD loop deletion analysis, the lack of ability to identify an inhibitory α-LptD mAb suggests that LptD may be specifically evolved to withstand extracellular modulation by antibodies while protecting critical structural elements of LptD function.

## Discussion

The OM is a critical feature of Gram-negative bacteria and presents a barrier to the discovery of new antibiotics (*Lewis, 2013*). There are two essential OMPs embedded in the OM and involved in construction of the OM barrier. BamA is part of the β-barrel assembly machine that folds and inserts β-barrel OMPs, including BamA and LptD, into the OM (*Konovalova et al., 2017*; *Ricci and Silhavy, 2012*). LptD is the last protein in the Lpt pathway that establishes and maintains OM asymmetry by inserting LPS exclusively into the outer leaflet of the OM (*Okuda et al., 2016*). Despite the importance of these β-barrel OMPs, understanding of how their structures contributes to their folding and function is still incomplete.

Similar to other Gram-negative β-barrel OMPs, both BamA and LptD possess short periplasmic loops but long, variable, and environmentally-exposed ECLs (*Franklin and Slusky, 2018*; *Rollauer et al., 2015*; *Schulz, 2002*). A deeper examination of the roles of these ECLs in a native OM context could contribute to a greater understanding of how they contribute to the structure and function of OMPs in general. To systematically interrogate the LptD ECLs, we performed four antibody discovery campaigns to generate thousands of antibodies targeting LptD to probe structure-function relationships of this essential OMP in its native OM environment. The unusual breadth and depth of our approach represents a powerful way to probe the accessible surfaces of the ECLs of LptD and to tease apart the changes that accompany LPS insertion by LptD in the native OM environment of the bacterial cell. By characterizing hundreds of these α-LptD mAbs, we discovered that extensive portions of the extracellular surface of LptD are dispensable in *E. coli* and critical features of LptD are likely occluded and protected by non-essential loops (*Figure 5*).

Little is known about the role and dynamics of the ECLs of LptD in the function of LPS insertion into the OM and there are few available tools to study this process. In an effort to probe as many extracellular epitopes as possible, we generated a large, diverse α-LptD mAb library by undertaking multiple immunization campaigns utilizing different antigens, adjuvants, and hosts. Varying antigens systematically presented LptD in increasingly complex arrangements to potentially capture different available epitopes. First, peptide immunizations utilized both linear peptides derived from conserved and non-conserved LptD ECL sequences and cyclic variants in an effort to constrain particular conformations. Next, purified full-length LptDE immunizations aimed to capture a more native state of the ECLs. Because the matrix in which the protein is reconstituted can alter the conformational flexibility

**Table 2.** mAbs to accessible ECLs do not inhibit essential function of LptD.

| Antigen | Host | Clones | Growth inhibitory α-LptD mAbs* | | |
| --- | --- | --- | --- | --- | --- |
| | | | K-12[†] | ΔwaaD[‡] | ΔwaaD + Rif.[§] |
| Linear and cyclic LptD peptides | SD Rats | 576 | 0 | 0 | 0 |
| Purified LptDE | Mice | 3360 | 0 | 0 | 0 |
| Purified LptDE | SD Rats | 2400 | 0 | 0 | 0 |
| Cells and purified LptDE | SD Rats | 1494 | 0 | 0 | 0 |

*For each antibody campaign, bacterial growth was measured (OD$_{600}$) for *E. coli* ΔwaaD and *E. coli* K-12 after treatment with each antibody at 10 µg/mL for 4 hr. Growth inhibition was calculated as a percentage of growth compared to an untreated control. 50% growth inhibition was considered positive.

[†]WT (wild-type) is *E. coli* K-12.

[‡]ΔwaaD is an *E. coli* ΔwaaD.

[§]ΔwaaD + Rif. is *E. coli* ΔwaaD grown in the presence sub-inhibitory rifampicin

DOI: https://doi.org/10.7554/eLife.46258.013

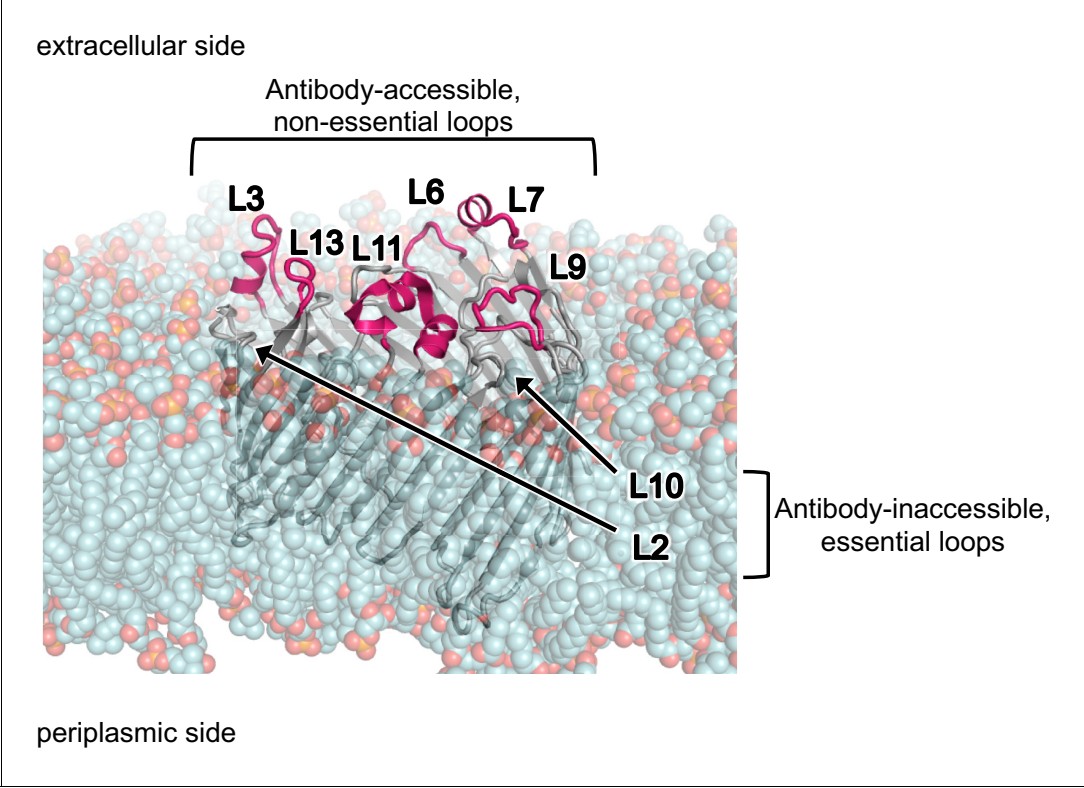

**Figure 5.** Antibody accessible ECLs of LptD. Side view of LptD structure rendered in PyMol from *E. coli* LptDE with LptE removed (PDB 4RHB [*Malojcic et al., 2015*]). LptD was placed in a standard phosphatidylethanolamine membrane context (shown as cyan and red space fill) using MemProtMD database (*Stansfeld et al., 2015*). Antibody accessible, non-essential ECLs L3, L6, L7, L9, L11, and L13 are highlighted in pink and antibody inaccessible ECLs L2 and L10 are indicated.

DOI: https://doi.org/10.7554/eLife.46258.014

The following figure supplement is available for figure 5:

**Figure supplement 1.** Diversity of the ECLs of LptD.

DOI: https://doi.org/10.7554/eLife.46258.015

and exposure of membrane-proximal epitopes, we utilized multiple reconstitution matrices (i.e., detergents and amphipols). Finally, using an approach that we have recently leveraged to find rare, inhibitory α-BamA antibodies, a targeted boost-and-sort strategy, we immunized rats with sub-lethal *E. coli* infections and then boosted them with purified LptDE protein to enrich for any LptD antibody response (*Vij et al., 2018*). Taken together, our effort to maximize the epitope coverage in our α-LptD mAb library led to distinct clusters of antibodies that possessed differing abilities to cross-react with LptD species variants, required different ECLs for binding, and exhibited varying levels of access to LptD through the protective LPS layer. Although the full breadth of antibody diversity in our library required the multiple immunization approaches described here, we propose a streamlined workflow for future OMP efforts. An accessible approach is to immunize SD rats with recombinant purified OMP either with or without whole bacterial cells (strategies 3 and 4, *Figure 1*). Both of these workflows returned the highest percentages of FACS+ antibodies coupled with recognition of diverse epitopes (*Figure 2*, *Figure 4B*, and *Figure 4—figure supplement 1*). Given the ease of manipulating bacterial strains, it could even be possible to steer immunizations towards particular ECLs by initially immunizing with a ΔECL-OMP strain followed by boosting with recombinant protein.

Other scaffolds or antibodies from different host species might be able to recognize distinct epitopes on the exposed portions of LptD. For example, recognition of partially buried epitopes could be achieved using species, such as rabbits, chickens, llamas, or camels, that contain longer CDRH3s that are better suited to reach into cavities on a particular antigen (*Lavinder et al., 2014*;

*Muyldermans and Smider, 2016*). In the case of the mouse and rat antibodies described here, we found that seven different LptD ECLs, out of a total of 13, could affect binding of the mAbs and the remaining six loops are likely partially or fully buried based on models from crystal structures (*Botos et al., 2016*; *Dong et al., 2014*; *Qiao et al., 2014*) (*Figure 5*). When this α-LptD mAb campaign is combined with of our α-BamA mAb effort (*Storek et al., 2018*; *Vij et al., 2018*), we now have the ability to probe the structure-function relationships of the only two essential OMPs in their native OM environment in *E. coli* at a level not previously possible. Importantly, we note the antibody discovery methods that we describe herein are efficient, robust, and a streamlined approach can be readily applied within a standard laboratory setting.

We have observed that in LptD there appear to be differential requirements for particular ECLs, L2 and L10, depending on the structure of LPS, specifically wild-type LPS versus the truncated *ΔwaaD* LPS. Notably, BamA appears to be defective in the highly fluid OM of an LPS-truncated strain grown under low salt and high temperature conditions (*Storek et al., 2018*). This is manifest by an increased sensitivity to a recently discovered inhibitory α-BamA antibody MAB1 and a requirement for elevated BamA levels. In these cases, the defects can be suppressed by altering the conditions to increase membrane rigidity (*Storek et al., 2019*). The LPS-dependent requirement for LptD loops suggests that effects of the OM structure on OMPs might be more general. Among the 13 extracellular LptD loops, both L2 and L10 are short and lack large insertions or deletions across LptD homologs when compared to the longer loops in highly divergent LptD proteins (*Figure 5—figure supplement 1A*). One possible reason for this is that these loops are uniquely positioned to make interactions with the LPS in the OM to allow for proper folding or functioning of this essential OMP. Although these interpretations are speculative and complicated by the fact that LPS is the substrate for LptD, it is tempting to hypothesize that the structure and function of OMPs evolved to be maximal when embedded in the constrained, liquid crystalline, OM matrix (*Paracini et al., 2018*) and altering the nature of this membrane leads to defects. Thus, membrane environment is a critical consideration when interpreting biochemical and structural data of LptD and other OMPs in detergent micelles and must be considered when attempting to design or discover inhibitors of these essential processes in Gram-negative bacteria.

Genetic, biochemical, and structural data support a model for LPS insertion into the OM whereby the acyl chains of lipid A are inserted into the hydrophobic bilayer through a lateral gate created between the first and last strands of the LptD β-barrel (*Botos et al., 2016*; *Dong et al., 2014*; *Qiao et al., 2014*). It is less clear how the polar sugars of the core oligosaccharide and O-antigen emerge from the periplasm to the cell exterior as it would be unfavorable for them to pass directly through the membrane. Presumably the sugars must pass through the LptD transmembrane β-barrel lumen, which is also occupied by an essential lipoprotein, LptE, and then through a proposed opening in the surface-exposed cap of LptD. Thus, consistent with prior molecular dynamic simulations, there are likely major structural surface re-arrangements in LptD during LPS insertion (*Dong et al., 2014*). Given this model, we hypothesized that an antibody could have interfered with this process in a number of ways. For example, a mAb could staple the lateral gate in either a closed or open conformation, prevent the movements associated with expelling the sugars to the outside, or allosterically alter the protein so that LPS movement is thwarted. It is informative, therefore, that none of our identified α-LptD mAbs were able to inhibit the activity of LptD. One explanation is that binding to a single epitope is insufficient to disrupt the required movements of LptD. Alternatively, the positions on LptD critical for these presumed structural changes might be inaccessible to mAbs. Indeed, few or no α-LptD mAbs bound to the extracellular LptD loops that were found genetically to be important for viability or OM maintenance under specific conditions. Consistent with this possibility, in crystal structures of LptDE, the two loops we found most critical were occluded by other parts of the protein: L2 by L3, and L10 by L9 and L11 (*Dong et al., 2014*; *Qiao et al., 2014*).

By burying critical regions, LptD appears to avoid direct interference by antibodies, providing bacteria an evolutionary advantage in evading functional antagonism by the host immune system. Accordingly, the antibody-accessible LptD loops also are the most diverse in sequence, suggesting that these regions may be subject to selective pressures from the immune system (*Botos et al., 2016*) (*Figure 5—figure supplement 1B*). Moreover, the proximity of a particular epitope to the membrane can sterically or electrostatically block binding. However, in rare cases, recognition can be achieved via stabilizing hydrophobic residues in an antibody CDR that transiently interact with the membrane, as observed for several α-HIV or α-GPCR antibodies (*Ishchenko et al., 2017*;

Scherer et al., 2010). It remains to be seen whether immobilizing these loops in LptD, for example, with a small molecule or peptide macrocycle that can access these positions, would inhibit LptD activity. While our observations make direct interference with LptD function by antibodies unlikely, they do not rule out the potential value of LptD as a vaccine candidate (Zha et al., 2016).

Overall, our study represents an exhaustive α-LptD antibody discovery campaign, and potentially the most exhaustive on any membrane protein described to date. For the first time, we have been able to map essential and non-essential structural features of this intriguing and important potential antibiotic target. We have observed extreme tolerability of the ECLs to interference, suggesting that the architecture of the exposed ECLs of LptD play a role in protecting critical functions of this essential OMP. More generally, we provide a robust template for future efforts to dissect the structure-function relationships of complex membrane proteins from bacteria and mammals in their native environments, and a method to rapidly assess the therapeutic potential of antibody targeting.

## Materials and methods

### Growth conditions
Luria-Bertani broth (ThermoFisher Scientific 12795027) and Mueller Hinton II cation-adjusted broth (BBL 212322) was prepared according to manufacturer's instructions. Bacterial cultures were grown at 37˚C. When appropriate, media was supplemented with kanamycin (50 µg/mL), carbenicillin (50 µg/mL), chloramphenicol (12.5 µg/mL), hygromycin (200 µg/mL), gentamicin (10 µg/mL) and arabinose (0.2% vol/vol).

### Bacterial strains and plasmids
Bacterial strains and relevant primers are listed in *Supplementary file 2*. Kanamycin deletion-insertion mutations of *waaD was* obtained from the Keio collection (*Baba et al., 2006*). Briefly, pKD4 or pKD3 was amplified with primers containing 50 bp nucleotide homology extensions (*Supplementary file 2*) to the gene of interest. The linear product was transformed into the appropriate background strain containing pSIM18 (*Chan et al., 2007*) recovered for 4 hr and selected on media containing 50 µg/mL kanamycin or 12.5 µg/mL chloramphenicol as appropriate. Mutations were confirmed by PCR. To make sequential mutants, the antibiotic marker was flipped out using pCP20 (*Datsenko and Wanner, 2000*).

The conditional *lptD* strain, Δ*lptD*::P$_{BAD}$-*lptD*, was created by inserting P$_{BAD}$-*lptD* at the *attB* site in MG1655 followed by deletion of the native copy of *lptD* (*Datsenko and Wanner, 2000*; *Diederich et al., 1992*). Briefly, *lptD* was cloned into pBAD24 using standard methods. P$_{BAD}$-*lptD* was amplified from pBAD24-*lptD* and sub-cloned into pLDR9. pLDR9-P$_{BAD}$-*lptD* was digested with XbaI, ligated, and transformed into MG1655 expressing pLDR8. PCR and DNA sequencing confirmed insertion of P$_{BAD}$-*lptD* at the *attB* site. After integration of P$_{BAD}$-*lptD*, the native copy of *lptD* was deleted using λ Red recombination as described above. In the absence of arabinose, the conditional D*lptD*::P$_{BAD}$-*lptD* strain did not grow. To obtain a conditional *lptD* strain truncated for LPS, *waaD* was deleted using λ Red recombination.

To clone the loop mutants into the *lptD* conditional strains, pLMG18 was modified to replace the original antibiotic cassette providing chloramphenicol resistance with a gentamicin resistance cassette using Gibson Assembly. The *lptD* gene and mutant constructs were codon optimized and ordered as gBlocks (Integrated DNA technologies). The genes were amplified and cloned into pLMG18gm using Gibson Assembly. Clones were confirmed by sequencing.

### Expression and purification of *E. coli* LptDE
The *E. coli* LptDE expression plasmid was constructed by cloning *E. coli* LptD (amino acids 1–784) and *E. coli* LptE (amino acids 1–193) into pCDFDuet1 vector at the first and second multiple cloning sites. An AviTag followed by an 8x histidine tag was inserted into N-terminal of *E. coli* LptD. The expression plasmid was expressed in *E. coli* C41(DE3). When the OD$_{600}$ reached 0.9, cells were induced with 0.5 mM isopropyl-β-D-1 thiogalactopyranoside (IPTG) overnight at 18˚C. Cells were harvested by centrifugation at 4500 rpm for 15 min and the cell pellets were resuspended in lysis buffer (50 mM Tris, pH 8.0, 200 mM NaCl, 1x complete protease inhibitor mixture (Roche)). The resuspended cells were disrupted by passing through a microfluidizer at 10,000 psi three times. The

cell lysate was added with 2% Zwittergent 3–14 and rocked overnight at 4°C. The suspension was ultracentrifuged (45Ti rotor, 40,000 rpm) for 1 hr at 4°C. The supernatant was incubated with pre-equilibrated Ni-NTA resins (Qiagen) and rocked for 1 hr at 4°C. The resins were washed by the buffer containing 50 mM Tris, pH 8.0, 200 mM NaCl, 1% Zwittergent 3–14, 15 mM imidazole, 1x complete protease inhibitor mixture and eluted with the buffer consisting of 50 mM Tris, pH 8.0, 200 mM NaCl, 1% Zwittergent 3–14, 300 mM imidazole and 1x complete protease inhibitor mixture. The eluate was concentrated and purified by a Superdex 200 16/60 column (GE Healthcare) using 20 mM HEPES, pH 8.0, 100 mM NaCl, 1.5% n-Octyl-β-D-Glucopyranoside (OG; Anatrace), 1x complete protease inhibitor mixture as the running buffer. The obtained LptD/E protein was diluted 4-fold into the buffer of 20 mM HEPES, pH 8.0, 1.5% OG and further polished using Q HP anion exchange 5 ml column (GE Healthcare). 20 mM HEPES, pH 8.0, 25 mM NaCl, 1.5% OG was used as the start buffer and elution was achieved with linear gradient of 0.025–1 M NaCl. When protein was required in amphipol, LptD/E protein was incubated with biotinylated amphipol (Anatrace) overnight at 4°C and then purified over a Superdex 200 16/60 column in a buffer consisting of 20 mM HEPES, pH 8.0, 100 mM NaCl.

## Peptide pools
Peptide pools are listed in *Supplementary file 3*. Peptides were designed based on sequence conservation, surface exposure and loop location.

## Protein biotinylation, amphipol reconstitution, and PE-labeling
When LptDE protein was required for ELISA or antigen-based sorting, an in vitro biotinylation reaction using BirA enzyme targeting the N-terminal Avi tag was first carried out according to the manufacturer's suggestions (Avidity); LptDE was then rerun over the Superdex 200 (26/60) in buffer E as described above in order to remove free biotin and other reaction components. When protein was required in amphipol (either non-biotinylated or biotinylated), LptDE at 1 mg/mL in buffer E was incubated with a stock solution of 2 mg/mL A8-35 amphipol (Anatrace) at 22°C for 1 hr and then applied over a Superdex 200 (26/60) column in buffer F (50 mM Tris pH 8, 100 mM NaCl). Protein reconstituted in A8-35 amphipol for immunization was concentrated to 1 mg/mL using a centrifugal device (10 K MWKO; Millipore). When PE-labeled protein was required for antigen-specific sorting, LptDE at 1 mg/mL (biotinyated and reconstituted in A8-35 amphipol in buffer F) was mixed 1:1 (vol/vol) with PE-streptavidin (Jackson ImmunoResearch) reconstituted in buffer F; the PE-streptavidin-biotin-LptDE-amphipol complex was incubated at 22°C for at least 15 min prior to use.

## Generation and purification of hybridomas
All animal study designs for the mouse and rat immunizations were reviewed and approved by the Genentech Institutional Animal Care and Use Committee prior to the start of this work. All animal work was performed in accordance with relevant guidelines and regulations.

## Generation of murine hybridomas
LptD peptide immunization campaign immunized Balb/c mice (Charles River Laboratories, Hollister, CA) with cyclic and linear LptD peptide pools (*Supplementary file 3*). The initial immunization contained 100 µl Complete Freund's Adjuvant (CFA) and subsequent dosing included 100 µl Incomplete Freund's Adjuvant (IFA) (Sigma). pAbs were purified by Protein A and assayed by ELISA, FACS, and inhibition of bacterial growth as described below. Hybridoma fusions were performed as previously described and supernatants were screened for protein binding by ELISA (*Hazen et al., 2014*). All ELISA positive clones were purified and screened for inhibition of bacterial growth.

LptDE protein immunization campaign immunized Balb/c mice (Charles River Laboratories, Hollister, CA) with detergent-solubilized or amphipol-reconstituted *E. coli* LptDE protein and either CFA or Ribi adjuvants (Sigma). pAbs were purified and screened as described above.

## Generation of rat hybridomas
LptDE protein immunization campaign immunized Sprague Dawley rats (Charles River Laboratories, Hollister, CA) with detergent-solubilized or amphipol-reconstituted *E. coli* LptDE protein and either CFA or Ribi adjuvants (Sigma). pAbs were purified and screened as described above.

Bacterial immunization campaign immunized Sprague Dawley rats (Charles River Laboratories, Hollister, CA) with *E. coli* K-12 cells in PBS ($10^7$–$10^9$ colony forming units via intravenous injection). The rats were boosted four times with *E. coli* K-12 cells followed by two protein boosts with 10 µg *E. coli* LptDE protein in amphipol. To generate monoclonal antibodies, hybridoma fusions were performed as previously described except with a myeloma partner SP2ab that enables surface display of IgG cell (Hazen et al., 2014; Price et al., 2009). After HAT selection in ClonaCell-HY Medium C (StemCell Technologies) for 4 days, hybridomas were stained with a cocktail of FITC-conjugated anti-rat IgG1/IgG2a/IgG2b mAbs (1:100 dilution; Bethyl Laboratories) and PE-conjugated LptDE protein antigen (5 µg/µl). Samples were sorted on a FACSAria II cell sorter (BD Biosciences). Gating strategy to identify hybridoma cells was first based on size (FSC/SSC). After dead-cell exclusion with Propidium iodide (Sigma-Aldrich P-4864), IgG +LptDE $^+$ single cell hybridomas were sorted into 96-well tissue-culture plates containing 200 µl of ClonaCell-HY Medium E (StemCell Technologies). Profiles were analyzed by FlowJo v.9.7.7 software. Sorted cells were cultured for seven days.

## Hybridoma culture and purification

Hybridomas were cultured using a previously described semi-automated high throughput process for hybridoma culturing and antibody purification (Vij et al., 2018).

## Epitope binning and affinity characterization

Epitope bins were determined by 96 × 96 array-based SPR imaging system (Carterra, USA) using classical sandwich method. Purified antibodies were diluted at 20 µg/ml in 10 mM sodium acetate buffer pH 4.5. Using amine coupling, antibodies were directly immobilized onto a SPR sensorprism CMD 200 M chip (XanTec Bioanalytics, Germany) using a Continuous Flow Microspotter (Carterra, USA) to create an array of 96 antibodies. For antibody binning, printed chip was loaded on IBIS MX96 SPRi (Carterra USA) and *E. coli* LptDE protein, diluted to 50 nM in 1.5% bOG HBS-P buffer, was injected over the chip at 25°C followed by a second injection of purified antibody, diluted at 20 µg/ml in 1.5% bOG HBS-P buffer, to make a sandwich. The epitope binning data were processed using Carterra binning software tool. Epitope binning experiments were performed as part of our high throughput antibody screening workflow and are purely descriptive.

## Antibody activity assay

The test strain was grown to log phase in MHB II supplemented with 0.002% Tween-80, diluted to a final $OD_{600}$ 0.01 in sterile round-bottom 96-well plates (Costar). Antibodies were added at 10 µg/mL, incubated statically at 37°C and monitored for bacterial growth after 4 hr of static incubation. A plate reader at $OD_{600}$ measured optical density of bacterial growth after shaking the plate for 25 s. Bacterial inhibition was calculated by subtracting the $OD_{600}$ value from the media control, followed by dividing that value by the no antibody control $OD_{600}$ value. Antibody activity experiments were performed as part of our high throughput antibody screening workflow and are purely descriptive.

## ELISA

Antibodies were screened by capture ELISA. Briefly, 50 µl of biotinylated LptDE protein, diluted in assay buffer (PBS + 1.5% OG +0.5% BSA) was added to streptavidin coated 384 well plates (Thermo Scientific USA, Cat # 15504) and incubated at RT for 1 hr, while shaking. The plates were washed 3X with wash buffer (PBS + 1.5% OG). 50 µl of supernatants or purified antibody, neat or diluted in assay buffer, were added to the wells and incubated at RT for 1 hr, while shaking. The plates were washed 3X with wash buffer. The captured antibody was detected with goat anti-rat HRP or goal anti-mouse HRP secondary antibody as appropriate (50 µl per well diluted at 1:10K in assay buffer, Bethyl Laboratories USA, Cat # A110-236P). The plates were incubated at RT for 1 hr, washed 3X with wash buffer and 50 µl of substrate, TMB solution (Surmodics, USA Cat # TMBW-1000), was added to each well. The plates were incubated for 5 min at RT followed by addition of 50 µl of TMB stop solution (Surmodics, USA Cat # LPSP-1000). Plates were read at 630 nm. ELISA assays were performed as part of our high throughput antibody screening workflow and are purely descriptive.

## Bacterial flow cytometry assays

Bacterial cells were grown to log phase, pelleted, re-suspended in ice-cold PBS with 1% BSA, and normalized to an $OD_{600}$ of 0.5. Bacterial suspensions were diluted 1:2.5 in ice-cold PBS, 1% BSA and $2 \times 10^6$ CFU were incubated with 10 µg/ml mAb at 4°C for 1 hr under gentle agitation. Bacterial cells were then washed three times in ice-cold PBS, 1% BSA and labeled using Alexa 488 anti-mouse IgG (H + L) (1:1000; Invitrogen) or Alexa 488 anti-rat IgG (H + L) (1:1000; Invitrogen) for 30 min 4°C. After three washes in ice-cold PBS, bacterial cells were fixed by adding one vol. of 2% w/v paraformaldehyde in PBS. Samples were run on a BD FACSCalibur and data from 10,000 events were analyzed using FlowJo software. FACS experiments were performed as part of our high-throughput antibody screening workflow and are purely descriptive. An isotype matched antibody was used as a negative control.

## Western blots

Bacterial cells were grown to log phase, normalized to $OD_{600}$, and pelleted. Samples were resuspended in 1x LDS sample buffer (ThermoFisher Scientific) and boiled for 5 min prior to loading on a 4–12% Bis-Tris SDS-PAGE gel. Proteins were transferred onto cellulose membranes using the iBlot 2 Dry Blotting System (ThermoFisher Scientific). Membranes were blocked for 1 hr in Blocking Buffer (TBS containing 5% nonfat milk and 0.05% Tween-20), washed, then incubated either overnight at 4°C or room temperature for 1 hr with the following primary Abs: human α-LptD 3D11 (1 µg/mL, Genentech) and rabbit α-GroEL (1:25,000, Enzo). Appropriate HRP-linked secondary antibodies (GE Healthcare) were diluted 1:20,000 in TBST and incubated with the membrane for 1 hr at RT. Blots were developed using ECL Prime Western Blotting Detection Reagent (Amersham). The displayed Western blot experiment shows a single biological replicate.

## Bacterial growth assays

Growth curves were performed in biological triplicate. Bacterial strains were grown overnight on LB agar containing gentamicin and arabinose. Cells were scraped from the plate into fresh media. $OD_{600}$ was measured and subsequently diluted to 0.001 ($OD_{600}$). 100 µl transferred to a 96-well plate (Corning) and monitored for growth by measuring $OD_{600}$ (EnVision Multimode Plate Reader, PerkinElmer).

Bacterial growth curves were analyzed by determining the doubling time (dt) during exponential growth phase and compared via the unpaired Student's $t$ test using Prism 6.0 (GraphPad Software) (*Supplementary file 1*). The Bonferroni correction was applied to control for multiple comparisons.

Minimum inhibitory concentration (MIC) assays were performed by inoculating $10^5$ colony forming units into medium with only test antibiotics (rifampicin and vancomycin). Biological duplicates were performed. MIC defined as the lowest concentration of drug to inhibit bacterial growth.

## Acknowledgements

We thank Marcy Auerbach, Avinash Gill, Sophia Lee, Peter Luan, Sy-Hyun Kim, Min Xu, Summer Park, Maikke Ohlson, Ashley Fouts, Jeremy Stinson, and members of the Infectious Diseases Department (Genentech) for their insight and feedback. This study was supported by internal Genentech funds.

## Additional information

### Competing interests

Kelly M Storek, Joyce Chan, Rajesh Vij, Nancy Chiang, Zhonghua Lin, Jack Bevers III, Christopher M Koth, Jean-Michel Vernes, Y Gloria Meng, JianPing Yin, Heidi Wallweber, Olivier Dalmas, Stephanie Shriver, Christine Tam, Kellen Schneider, Dhaya Seshasayee, Gerald Nakamura, Peter A Smith, Jian Payandeh, James T Koerber, Laetitia Comps-Agrar, Steven T Rutherford: Employee of Genentech, Inc, a member of the Roche Group, and shareholder in Roche.

## Funding

| Funder | Author |
| --- | --- |
| Genentech | Kelly M Storek |
| | Joyce Chan |
| | Rajesh Vij |
| | Nancy Chiang |
| | Zhonghua Lin |
| | Jack Bevers III |
| | Christopher M Koth |
| | Jean-Michel Vernes |
| | JianPing Yin |
| | Heidi Wallweber |
| | Olivier Dalmas |
| | Stephanie Shriver |
| | Christine Tam |
| | Kellen Schneider |
| | Dhaya Seshasayee |
| | Gerald Nakamura |
| | Peter A Smith |
| | Jian Payandeh |
| | James T Koerber |
| | Laetitia Comps-Agrar |
| | Steven T Rutherford |

The funders had no role in study design, data collection and interpretation, or the decision to submit the work for publication.

## Author contributions

Kelly M Storek, Conceptualization, Data curation, Validation, Investigation, Visualization, Methodology, Writing—original draft, Writing—review and editing; Joyce Chan, Data curation, Validation, Investigation, Methodology, Writing—review and editing; Rajesh Vij, Data curation, Validation, Investigation, Visualization, Methodology, Writing—review and editing; Nancy Chiang, Zhonghua Lin, Data curation, Investigation, Methodology; Jack Bevers III, Resources, Investigation, Visualization, Methodology; Christopher M Koth, Peter A Smith, Conceptualization, Supervision; Jean-Michel Vernes, Investigation, Methodology; Y Gloria Meng, Conceptualization, Supervision, Methodology; JianPing Yin, Resources, Investigation, Methodology; Heidi Wallweber, Olivier Dalmas, Kellen Schneider, Investigation; Stephanie Shriver, Christine Tam, Resources, Methodology; Dhaya Seshasayee, Conceptualization, Supervision, Investigation; Gerald Nakamura, Conceptualization, Supervision, Methodology, Writing—review and editing; Jian Payandeh, James T Koerber, Conceptualization, Resources, Data curation, Supervision, Visualization, Methodology, Project administration, Writing—review and editing; Laetitia Comps-Agrar, Conceptualization, Data curation, Supervision, Visualization, Methodology, Project administration, Writing—review and editing; Steven T Rutherford, Conceptualization, Supervision, Visualization, Methodology, Writing—original draft, Project administration, Writing—review and editing

## Author ORCIDs

Steven T Rutherford (iD) https://orcid.org/0000-0002-4758-4248

## Ethics

Animal experimentation: All animal study designs for the mouse and rat immunizations were reviewed and approved by the Genentech Institutional Animal Care and Use Committee prior to the start of this work. All animal work was performed in accordance with relevant guidelines and regulations.

## Decision letter and Author response

Decision letter https://doi.org/10.7554/eLife.46258.021
Author response https://doi.org/10.7554/eLife.46258.022

## Additional files

### Supplementary files

• Supplementary file 1. Statistical analysis of the initial growth rates for conditional *lptD* deletion strains complemented with *lptD* loop mutants. Bacterial growth curves in *Figure 3* and *Figure 3— figure supplement 1* were analyzed by determining the doubling time (dt) during exponential growth phase and compared via the unpaired Student's *t* test. The Bonferroni correction was applied to control for multiple comparisons.
DOI: https://doi.org/10.7554/eLife.46258.016

• Supplementary file 2. Strains, plasmids, and primers used in this study. Names, descriptions, and references for all key resources (bacterial strains, plasmid constructs, and primers) as described in the text.
DOI: https://doi.org/10.7554/eLife.46258.017

• Supplementary file 3. Sequences of linear and cyclic peptides used for immunizations. LptD peptides used for immunizations (*Figure 1*) were designed based on sequence conservation, surface exposure and loop location.
DOI: https://doi.org/10.7554/eLife.46258.018

• Transparent reporting form
DOI: https://doi.org/10.7554/eLife.46258.019

### Data availability

All data generated or analyzed during this study are included in the manuscript and supporting files.

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
