## [Decision Letter]

Thank you for submitting your article "Massive antibody discovery used to probe structure-function relationships of the essential outer membrane protein LptD" for consideration by *eLife*. Your article has been reviewed by three peer reviewers, including Volker Dötsch as the Reviewing Editor and Reviewer #1, and the evaluation has been overseen by Gisela Storz as the Senior Editor. The following individuals involved in review of your submission have agreed to reveal their identity: Edmund RS Kunji (Reviewer #3).

The reviewers have discussed the reviews with one another and the Reviewing Editor has drafted this decision to help you prepare a revised submission.

Summary:

Storek et al. describe an exhaustive immunisation strategy to assess the use of an antibody-based approach as an antimicrobial agent against one of the essential OMPs in *E. coli*, LptD. They find that only a small fraction of mAbs (FACS^+^) were successful in *E. coli* K12 with intact LPS and were much more successful at raising antibodies in the *E. coli* Δ*waaD* conditional strain with a chromosomal arabinose-inducible LptD that produces a truncated from of LPS. Many of the mAbs recognised extracellular loops in LptD. However, by expressing many extracellular loop deletions they found out that all the FACS^+^ mAb bound to loops that were not required to function and had no antimicrobial effect. The most essential loops were L2 and L10, which are closer to the membrane. The authors did manage to produce mAb against L2 and L10 loops by using LptD-derived peptides for immunization instead of the full-length protein. Whilst these antibodies were not therapeutically useful in the *E. coli* K12 strain with an intact LPS, they nonetheless provided some interesting insights into the different roles of the loops. The authors concluded that the more accessible and longer extracellular loops might be "decoys" to avoid the immune system to produce antibodies against the inner loops L2 and L10 that were essential for function.

Essential revisions:

1) Subsection “Identifying α-LptD mAbs that bind to extracellular LptD loops”: "…while <1% were FACS^+^ on *E. coli* K-12 (Figure 2A)…"

What were the properties of these antibodies, as they can recognise LptD despite there being a lipopolysaccharide layer? Were they false positives? Which species and immunization protocols were used?

2) Subsection “Identifying α-LptD mAbs that bind to extracellular LptD loops” and onwards: "Indeed, we identified 10 (7%) α-LptD mAbs that bound to LptDE from all three species and 42 that bound either *E. coli* plus *K. pneumoniae* (22%) or *E. coli* plus *E. cloacae* (9%) (Figure 2C)."

Could the loop replacement approach (Figure 3B and related text) be used to confirm the location of the cross-reactive loops? See also Figure 4B as a strategy. This is relevant because it could show a common or differential bacterial strategy to evade an immune response and would firm up the claim that there is an evolutionary link. I find Figure 2D not very informative (unless you have the particular antibodies). More relevant would be to understand where the cross-reactive loops are. Maybe better to map the cross-reactive epitopes on the structure, based on the experimental results?

3) Subsection “Effects of removing extracellular LptD loop on *E. coli* growth” and onwards. "These results suggest that extracellular L2, L4, L8 and L10 are important."

Maybe another remarkable result is that the shortest loops cannot be easily deleted, whereas the longer loops are fairly tolerant (with the possible exception of loop 8 which was partially deleted). What it may tell you is that, structurally, the short loops are possibly as short as they can get to be compatible with a well-folded and therefore functional protein.

4) Subsection “Effects of removing extracellular LptD loop on *E. coli* growth” and Figure 3—figure supplement 1A).

Where is the expression profile of LptDΔL2? Legend says that it could not be evaluated, but maybe LptDΔL2is not produced, explaining the severe phenotype, as we know the protein is essential.

5) Have these OMPs evolved extracellular loops as "decoys" or these extracellular loops just happen to pose a technical issue when using an antibody-based approach to target OMPs? Is there any other data to support this conclusion, i.e., do non-pathogenic strains harbouring similar OMPs have shorter extracellular loops?

---

## [Author Response]

Essential revisions:1) Subsection “Identifying α-LptD mAbs that bind to extracellular LptD loops”: "…while <1% were FACS^+^ on E. coli K-12 (Figure 2A)…"What were the properties of these antibodies, as they can recognise LptD despite there being a lipopolysaccharide layer? Were they false positives? Which species and immunization protocols were used?

We provided additional information about the K-12 FACS^+^ antibodies in the text and have added an new data figure to provide more insight. First, the K-12 FACS^+^ antibodies were all LptDE ELISA^+^ (subsection “Identifying α-LptD mAbs that bind to extracellular LptD loops”) indicating they bound the protein. Second, their discovery was not limited to any one particular immunization campaign (subsection “Identifying α-LptD mAbs that bind to extracellular LptD loops”). Third, binding to LptD on *E. coli* K-12 cells was indistinguishable from binding to LptD on the *E. coli* Δ*waaD* cells (subsection “Identifying α-LptD mAbs that bind to extracellular LptD loops”and new Figure 2—figure supplement 1A), suggesting the epitope is not altered between the strains but rather extends further from the cell surface past the K-12 LPS layer. We also analyzed a representative *E. coli* K-12 FACS^+^ mAb for binding to our LptD loop mutant panel and report the binding profile for comparison to the *E. coli* Δ*waaD* binders (subsection “Mapping surface loop binding sites of α-LptD antibodies” and new Figure 2—figure supplement 1B).

2) Subsection “Identifying α-LptD mAbs that bind to extracellular LptD loops”and onwards: "Indeed, we identified 10 (7%) α-LptD mAbs that bound to LptDE from all three species and 42 that bound either E. coli plus K. pneumoniae (22%) or E. coli plus E. cloacae (9%) (Figure 2C)."Could the loop replacement approach (Figure 3B and related text) be used to confirm the location of the cross-reactive loops? See also Figure 4B as a strategy. This is relevant because it could show a common or differential bacterial strategy to evade an immune response and would firm up the claim that there is an evolutionary link. I find Figure 2D not very informative (unless you have the particular antibodies). More relevant would be to understand where the cross-reactive loops are. Maybe better to map the cross-reactive epitopes on the structure, based on the experimental results?

Yes, this important information can be obtained from our reported data and we have altered our figures and added text to address these relevant questions. The data pertaining to binding by cross-reactive antibodies to LptD loop mutants is captured across Figure 2, Figure 4 and Figure 4—figure supplement 1. To highlight the cross-reactive antibodies more, we have (1) swapped in an additional representative cross-reactive antibody (1H12) into the highlighted antibodies shown in Figure 4B to show its distinct binding pattern and (2) altered Figure 4—figure supplement 1 to indicate which antibodies bind only *E. coli* LptDE (hatched bars) and which bind LptDE from 2 or more species (solid bars). These comparisons are further resolved in Figure 2D. We have also highlighted each of the representatives shown in Figure 4B by notating those in Figure 2D. As these changes should make it easier to relate Figure 4 and Figure 4—figure supplement 1 to Figure 2D, we think that this makes Figure 2D more relevant as well and have kept it in the figure. We have also pointed the reader to Figure 2B and Figure 2—figure supplement 2 to provide a structural context for the conserved regions of LptD. These additions and changes have made some important trends much more apparent for the reader and are now described in the text (subsection “Mapping surface loop binding sites of α-LptD antibodies”). Briefly, a majority of the two- or three-species cross-reactive α-LptD mAbs were dependent upon L8 and L9 for binding. The cross-reactive mAbs (5A3 and 3D4) recognize slightly different epitopes comprised of L6/8/9/11 and L7/8, respectively, consistent with our in vitro epitope binning experiment.

3) Subsection “Effects of removing extracellular LptD loop on E. coli growth” and onwards. "These results suggest that extracellular L2, L4, L8 and L10 are important."Maybe another remarkable result is that the shortest loops cannot be easily deleted, whereas the longer loops are fairly tolerant (with the possible exception of loop 8 which was partially deleted). What it may tell you is that, structurally, the short loops are possibly as short as they can get to be compatible with a well-folded and therefore functional protein.

Although we have not directly shown this, we agree that this is a viable hypothesis. We have now included a short discussion point highlighting this idea (Discussion section). Additionally, to provide more context, we have also included an alignment of the *Enterobacteriaceae* LptD protein with even more divergent LptD species in a new figure (Figure 5—figure supplement 1A) that further highlights the plasticity in length of the ‘long’ loops compared to the ‘short’ loops. This comparison is in line with the hypothesis that the long loops are more tolerant to perturbations.

4) Subsection “Effects of removing extracellular LptD loop on E. coli growth” and Figure 3—figure supplement 1A).Where is the expression profile of LptDΔL2? Legend says that it could not be evaluated, but maybe LptDΔL2 is not produced, explaining the severe phenotype, as we know the protein is essential.

Because LptD is essential, depletion of the wild-type copy renders the strain non-viable. If the production of an LptD variant does not support growth, expression levels of the LptD variant protein cannot be assessed. This was the case for L2. The expression profiles shown in Figure 3—figure supplement 1A measured LptD levels from K-12 *E. coli* grown in liquid culture. In this condition, the L2 mutant does not complement the conditional strain, thus this strain did not grow (Figure 3B) and LptD levels could not be measured. To determine if L2 is produced in a condition where it can support growth sufficiently (agar plate-grown *E. coli* Δ*waaD*), we measured LptD levels and have included the expression profile in a new figure (Figure 3—figure supplement 1B) demonstrating L2 is expressed near WT levels under that condition. This was not clear as described in the original version of the text made thus, we have modified the text (subsection “Effects of removing extracellular LptD loop on *E. coli* growth” and subsection “Mapping surface loop binding sites of α-LptD antibodies”) and figure legend to remove any confusion.

5) Have these OMPs evolved extracellular loops as "decoys" or these extracellular loops just happen to pose a technical issue when using an antibody-based approach to target OMPs? Is there any other data to support this conclusion, i.e., do non-pathogenic strains harbouring similar OMPs have shorter extracellular loops?

We employed our antibody-based approach because it would be the predominant selective pressure from the host’s adaptive immune system. Pathogenic or not, the host would still recognize both bacteria as foreign and mount an immune response against the bacteria, including the OMPs. Although we cannot make a definitive evolutionary argument at this time, there are additional data to support the ideas that the loops are under evolutionary pressure. To showcase this point, we have added an additional figure (Figure 5—figure supplement 1).

In Figure 5—figure supplement 1A, mentioned above, LptD loop alignments of the three closely-related *Enterobacteriaceae* species with 2 pathogens of high interest, *Pseudomonas aeruginosa* and *Acinetobacter baumannii* highlight the large variation in sequence diversity and loop lengths. In Figure 5—figure supplement 1B, the conservation of each amino acid from 150 LptD homologs ranging from 35-95% identity with *E. coli* LptD is mapped on the surface of the LptD structure. This figure strikingly illustrates that while the inside of the β-barrel is conserved, the extracellular loops are highly divergent. These points are discussed in the revised text (Discussion section).